# Computational Based Designing of a Multi-Epitopes Vaccine against *Burkholderia mallei*

**DOI:** 10.3390/vaccines10101580

**Published:** 2022-09-21

**Authors:** Muhammad Irfan, Saifullah Khan, Alaa R. Hameed, Alhanouf I. Al-Harbi, Syed Ainul Abideen, Saba Ismail, Asad Ullah, Sumra Wajid Abbasi, Sajjad Ahmad

**Affiliations:** 1Department of Oral Biology, College of Dentistry, University of Florida, Gainesville, FL 32611, USA; 2Institute of Biotechnology and Microbiology, Bacha Khan University, Charsadda 24461, Pakistan; 3Department of Medical Laboratory Techniques, School of Life Sciences, Dijlah University College, Baghdad 00964, Iraq; 4Department of Medical Laboratory, College of Applied Medical Sciences, Taibah University, Yanbu 41477, Saudi Arabia; 5School of Biomedical Engineering, Shanghai Jiao Tong University, Shanghai 200000, China; 6Department of Biological Sciences, National University of Medical Sciences, Rawalpindi 46000, Pakistan; 7Department of Health and Biological Sciences, Abasyn University, Peshawar 25000, Pakistan

**Keywords:** *Burkholderia mallei*, multi-epitopes vaccine, molecular dynamics simulation, TLR-4

## Abstract

The emergence of antibiotic resistance in bacterial species is a major threat to public health and has resulted in high mortality as well as high health care costs. *Burkholderia mallei* is one of the etiological agents of health care-associated infections. As no licensed vaccine is available against the pathogen herein, using reverse vaccinology, bioinformatics, and immunoinformatics approaches, a multi-epitope-based vaccine against *B. mallei* was designed. In completely sequenced proteomes of *B. mallei*, 18,405 core, 3671 non-redundant, and 14,734 redundant proteins were predicted. Among the 3671 non-redundant proteins, 3 proteins were predicted in the extracellular matrix, 11 were predicted as outer membrane proteins, and 11 proteins were predicted in the periplasmic membrane. Only two proteins, type VI secretion system tube protein (Hcp) and type IV pilus secretin proteins, were selected for epitope prediction. Six epitopes, EAMPERMPAA, RSSPPAAGA, DNRPISINL, RQRFDAHAR, AERERQRFDA, and HARAAQLEPL, were shortlisted for multi-epitopes vaccine design. The predicted epitopes were linked to each other via a specific GPGPG linker and the epitopes peptide was then linked to an adjuvant molecule through an EAAAK linker to make the designed vaccine more immunologically potent. The designed vaccine was also found to have favorable physicochemical properties with a low molecular weight and fewer transmembrane helices. Molecular docking studies revealed vaccine construct stable binding with MHC-I, MHC-II, and TLR-4 with energy scores of −944.1 kcal/mol, −975.5 kcal/mol, and −1067.3 kcal/mol, respectively. Molecular dynamic simulation assay noticed stable dynamics of the docked vaccine-receptors complexes and no drastic changes were observed. Binding free energies estimation revealed a net value of −283.74 kcal/mol for the vaccine-MHC-I complex, −296.88 kcal/mol for the vaccine-MHC-II complex, and −586.38 kcal/mol for the vaccine-TLR-4 complex. These findings validate that the designed vaccine construct showed promising ability in terms of binding to immune receptors and may be capable of eliciting strong immune responses once administered to the host. Further evidence from experimentations in mice models is required to validate real immune protection of the designed vaccine construct against *B. mallei*.

## 1. Introduction

Antibiotic resistance by bacterial pathogens is an emerging public health problem that affects medical care all over the world. Antibiotics play an important role in the fight against infectious diseases. In the last few decades, antimicrobials have been misused, which has resulted in the evolution of novel resistance mechanisms and increased spread of multi-drug resistance pathogens [1]. Among antibiotic resistant pathogens, *Burkholderia mallei* is an emerging bacterial pathogen that needs immediate attention [2]. *B. mallei* is a Gram-negative, pathogenic bacterium belonging to the *Burkholderiaceae* family. Morphologically it is a bipolar and obligatory aerobe [3]. It is considered a sub-species of *B. pseudomallei* due to its genome homology as determined by multi-locus sequence typing*. B. pseudomallei* is the causative agent of melioidosis, which is a severe infection of humans and animals that targets the skin and cause acute septicemia. *B. mallei* causes glanders in horses, mules, and donkeys; these solipeds also serve as a reservoir for transmission of infection into humans [4]. Glanders transmission occurs by direct and indirect contact with fomites. Humans are also infected by close contact with infected animals or laboratory cultures. Zoonotic infection is characterized by septicemia, chronic infection of the skin, and pneumonia [5]. *B. pseudomallei* and *B. mallei* show resistance to several antibiotics, and their mechanism of resistance has also been determined. *B. mallei* is mainly resistant to ciprofloxacin, co-amoxiclav, chloramphenicol, avibactam, and rifampin. The mechanism of ceftazidime resistance involves the loss of penicillin-binding protein and a mutation in genes that codes for beta-lactamase enzyme [6].

The immunity against *B. mallei* depends on the activation of innate immune response; the mechanism of activation of adaptive response against *B. mallei* remains unclear in glanders infection. The mechanism of the innate immune response against *B. mallei* includes both cellular and non-cellular or acute phase pathways [7]. Cellular pathways comprise intracellular ubiquitination and actin–cytoskeleton rearrangement. Signaling molecules of the immune system including interferon, tumor necrosis factor, and Toll-like receptors play vital roles in the activation of innate immune responses. Modifications in *B. mallei* lipopolysaccharide (LPS), particularly in the lipid A portion, induce immune responses via TLR-4 activations that assist in persistent infection [8].

A vaccine is an alternative way to reduce the spread of infectious disease. In several infectious cases, conventional vaccinology fails to make a good vaccine against pathogens that are unable to be grown in vitro [9]. A conventionally developed vaccine also lacks potency against antigenically variable strains [10]. Additionally, culture base vaccine formulation is very costly, time-consuming, and requires many laboratory resources [11]. In the recent past, bioinformatics and genomics have proved successful in the prediction of possible vaccine candidates in bacterial genomes [12]. Advances in bioinformatics, immunoinformatic, and reverse vaccinology pipelines are now commonly used in the area of vaccine candidate’s prioritization and vaccine designing [13]. Reverse vaccinology (RV) is a methodology employed for the prediction of good vaccine targets from a pathogen’s genome [14] and has been successfully used in the development of meningococcal serogroup B (4CMenB) [15]. Pan-genomic reverse vaccinology (PGRV) in particular is more efficient compared to pasture base vaccinology as PGRV predicts vaccine targets from the core proteome, which are considered promising broad-spectrum targets [16].

As *B. mallei* is an emerging bacterial pathogen and its antibiotic resistance pattern is expanding, urgent efforts are needed to devise new therapeutic strategies against this pathogen. Additionally, no vaccine is yet available which may further limit effective therapy against the bacteria. In this research, we applied RV and immunoinformatics approaches to design an in silico based multi-epitopes-based vaccine construct against *B. mallei*. In this process, the pathogen core proteome was identified, followed by prediction of potential vaccine proteins by considering multiple vaccine filters. The vaccine targets were then subjected to epitope mapping to predict potential B and T-cell epitopes. A chimeric vaccine containing multiple epitopes was then constructed. Biophysics techniques were then applied to the vaccine to understand its binding and dynamics behavior with immune receptors such as TLR-4, MHC-I, and MHC-II. The designed vaccine construct may be helpful for experimental scientists to develop an effective vaccine against *B. mallei*.

## 2. Research Methodology

The methods used for designing of a multi-epitope-based vaccine construct against *B. mallei* are schematically presented in Figure 1. 

### 2.1. Complete Proteome Extraction, BPGA Analysis, and Subtractive Proteomics Filters 

The study was initiated by retrieval of the complete proteome of the pathogen strains from the National Center of Biotechnology Information (NCBI) [17]. Then, bacterial pan-genome analysis (BPGA) (http://pgaweb.vlcc.cn/ (accessed on 2 March 2022)) was performed in order to retrieve core sequences. The core sequences were further processed with a redundancy check using the CD-HIT online webserver [18]. In CD-HIT analysis, all the duplicated proteins were removed and non-redundant proteins were further processed for subcellular localization analysis using the online webserver PSOSRTb [19]. After subcellular localization analysis, virulent factor data base (VFDB) analysis was performed using the VFDB online webserver [20]. Only proteins with bit score ≥100 and sequence identity ≥30% were chosen. Next, transmembrane helices were checked and those proteins with >1 transmembrane helices were discarded [21]. Next, antigenicity analysis was performed through the Vaxijen 2.0 webserver [22] considering a 0.6 threshold value. Only antigenic proteins were used in allergenicity and water solubility analysis. The allergenicity check was performed using AllerTOP 2.0 [23] while the solubility check was conducted through the InnovaGen 2.0 (https://pepcalc.com/peptide-solubility-calculator.php (accessed on 15 March 2022)) webserver. Furthermore, physicochemical analysis was evaluated using ProtParam Expasy online webserver [24]. Homology analysis was performed against Human proteome (taxonomic id: 9606) and probiotic bacteria such as *Lactobacillus rhamnosus* (taxonomic id: 47715), *L. johnsonii* (taxonomic id: 33959), and *L. casei* (taxonomic id: 1582). Homology proteins were discarded when showing identify of ≥30% [25]. This is vital to avoid auto-immune responses. 

### 2.2. Epitopes Selection Phase

In the epitopes selection phase, first B-cells epitopes were predicted using the IEDB online webserver [26]. Further, predicted B-cells epitopes were used for T-cells epitopes prediction [15]. Prediction of MHC-I and MHC-II alleles were predicted using a reference set of MHC molecules given at the IEDB server (http://tools.iedb.org/mhci/ (accessed on 15 March 2022) for MHC-I and http://tools.iedb.org/mhcii/ (accessed on 15 March 2022) for MHC-II). The predicted epitopes were further selected for antigenic probability test [27]. Additionally, allergic, less water soluble and toxic epitopes were filtered out using AllerTOP 2.0, InnovaGen (https://pepcalc.com/peptide-solubility-calculator.phpand (accessed on 15 March 2022)) and ToxinPred tool (http://crdd.osdd.net/raghava/toxinpred/ (accessed on 15 March 2022)), respectively. The final set of epitopes was considered for multi-epitopes vaccine construct design. Moreover, population coverage analysis was performed using IEDB server (https://www.iedb.org/ (accessed on 15 March 2022)). For comparative analysis, outer membrane protein A (ompA) of *Escherichia coli* was used as a positive control. The entry name of ompA in uniport is P0A910. OmpA is a potential antigen against the bacterial pathogen and has been extensively evaluated experimentally (doi.org/10.3389/fimmu.2020.01362 accessed on 15 March 2022). 

### 2.3. Multi-Epitopes Vaccine Construction Phase

Multi-epitopes vaccine construct was designed from selected epitopes [28]. The epitopes were connected to each other’s by the “GPGPG” linker and additionally linked to cholera toxin-B subunit adjuvant via another “EAAAK” linker to make the vaccine more immune potent [29]. Physicochemical properties of the designed vaccine were assessed by ProtParam online webserver [30]. The 3D structure was modeled through scratch predicted tool [31]. Moreover, the vaccine loops were refined using the refinement tool of the galaxyWeb webserver [32]. To further retain the structure stability, disulfide bonds were created by Design 2.0 online webserver [33]. Next, secondary structures and Ramachandran plot analysis were performed using PDBsum generate algorithm [34]. As stated above, OmpA was used as a positive control to cross-check the predictions made for the designed vaccine candidate. 

### 2.4. Molecular Docking Study

Interactions between the vaccine and immune cell receptors were evaluated through docking. In molecular docking, the binding efficiency of vaccine construct with immune cell receptors (MHC-I, MHC-II, and TLR-4) [35] was analyzed. Before docking, first we retrieved the immune receptor’s 3D structure from the Protein Data Bank (PDB) using a specific 4-digit code. Cluspro 2.0, which is an online docking software, was used for docking purposes [12,36]. The docking procedure was performed blindly, and the one with lowest energy score was selected for simulation studies. 

### 2.5. Molecular Dynamic Simulation 

The docked complexes with the least binding energy score were considered for molecular dynamic simulation analysis which was performed through AMBER20 software [37]. The simulation analysis was completed in three phases: pre-processing phase, simulation, and trajectories analysis phase [38]. Preprocessing of the complexes was conducted via the Antechamber program. The FF14Sb was used as a force field. Energy minimization was performed for 1500 steps using the steepest descent and conjugate gradient algorithms. The systems were heated up to 310 K, equilibrated, and simulated for 100 nanoseconds. Temperature control during the simulation was achieved using Langevin algorithm while hydrogen bonds were constrained through SHAKE algorithm. The output trajectories analysis consists of root-mean-square deviation (RMSD) [39] and root-mean-square fluctuation (RMSF) [40]. The simulation plots were generated through XMGRACE software (https://plasma-gate.weizmann.ac.il/Grace/ (accessed on 15 March 2022)). 

### 2.6. Binding Free Energies Estimation

Binding free energies were estimated for top-docked complexes through the MMGBSA approach. The net free binding energies estimation was performed to validate the docked stability of vaccine-immune receptor complexes. The lesser binding free energy describes a complex as more stable. A total number of hundred frames were investigated during MMGBSA analysis.

## 3. Results

### 3.1. Complete Proteome Extraction Phase and Bacterial Pan-Genome Analysis Phase

The study was commenced with the retrieval of complete five proteomes of the pathogen. The accession number of the pathogen strains are: ASM95958v1, ASM393301v1, ASM393302, ASM393303v1, and ASM393304v1. The strains have completely sequenced genomes and were subjected for bacterial BPGA analysis phase.

### 3.2. BPGA Phase and Subtractive Proteomics Filters 

BPGA predicted 18,405 core sequences. Core sequences offer a set of good broad-spectrum vaccine proteins as they are shared by all strains. The core–pan plot is mentioned in Figure 2. The core–pan plot demonstrates the number of gene families in each strain. The core sequences were subjected to redundancy analysis that predicted that the core sequences consist of 3671 non-redundant proteins and 14,734 redundant proteins. Non-redundant proteins have a single presentation in the proteomes and thus could save time and computational resources. The redundant proteins were discarded and the non-redundant proteins were further subjected to subcellular localization analysis. In subcellular localization analysis, 25 surface localized proteins were predicted of which 3 were extracellular proteins, 11 proteins were found in outer membrane region, and 11 were predicted in periplasmic membrane region. The surface proteins are good vaccine targets as they can be easily recognized by the host immune system. The subcellular localized proteins were further evaluated for virulence analysis. In extracellular membrane proteins, only two proteins were predicted as virulent while in outer membrane and periplasmic membrane proteins, six and two proteins, respectively, were found to have bit scores >100 and bit-score >30%. In total, 10 virulent proteins were predicted. The virulent proteins can stimulate infection and immune pathways and are considered good vaccine targets. The virulent proteins were further processed and non-virulent were discarded. Transmembrane helices were evaluated but no proteins were found to have more than one transmembrane helix. Low number of transmembrane helices proteins ensures easy experimental evaluation and can be clone and expressed readily. The proteins were further considered for antigenicity analysis and predicted five proteins as probable antigens with antigenicity scores of 0.98, 0.69, 0.61, 0.84, and 0.67. Antigenic proteins stimulate good immune reactions. The antigenic proteins were further processed for allergenicity analysis and predicted three proteins as an allergen. The allergen proteins were discarded and the non-allergen proteins were further processed. Water solubility, physicochemical properties analysis, and homology analysis were further conducted. In said analysis, no poor water soluble, physiochemically unstable, and similar proteins with host and host intestinal flora were found. The number of proteins filtered in each step is presented in Figure 3.

### 3.3. Epitopes Prediction and Prioritization Phase 

In epitopes prediction and prioritization phases, only two proteins; core/3507/1/Org1_Gene1451 (type VI secretion system tube protein (Hcp) and Query = core/426/1/Org1_Gene4503 (type IV pilus secretin PilQ) were shortlisted for the epitope selection phase. From the first protein only one epitope was predicted, and from the second protein five B-cell epitopes were predicted, as shown in Table 1.

### 3.4. T-Cells Epitopes Prediction 

In the T-cells epitopes prediction phase, both MHC-I epitopes and MHC-II epitopes were predicted. The selection of epitopes was based on lower percentile score. The predicted epitopes that were prioritized are mentioned in Table 2. The listed epitopes are B-cell derived T-cell epitopes, which can stimulate both humoral and cellular immunity at the same time. 

### 3.5. Epitopes Screening Phase 

Furthermore, the predicted epitopes were further screened for DRB*0101 binding affinity, antigenicity and allergenicity, water solubility, and toxicity. Only good DRB*0101 binders, probable antigenic, non-allergenic, and highly water-soluble epitopes were shortlisted for multi-epitope-vaccine designing. The DRB*0101 allele is the most abundant allele in humans and any antigen that binds to this allele has higher chances of presentation to the immune system and thus generates strong immunological reactions. Usually, epitopes with IC50 value less than 100 nM are considered strong binders. The shortlisted epitopes are tabulated in Table 3. For comparative purpose, ompA protein was used to cross-validate the predictions made for the epitopes. The ompA antigenic score is 0.6681; non-allergen, water soluble, and excellent DRB*0101 binder score is 0.86. 

### 3.6. Population Coverage Analysis

The selected epitopes were screened for population coverage analysis. This analysis revealed that the selected epitopes have the efficacy to cover 99.74% of world population. Countries wise, the vaccine has coverage of 97.83% of the Chinese population and 96.35% of the Indian population. Population coverage of the vaccine epitopes for different countries is shown in Figure 4. 

### 3.7. Multi-Epitopes Vaccine Construction and Processing

Multi-epitope-vaccine construct was designed so that the vaccine would have good immune potency compared to a single epitope vaccine. In multi-epitope-vaccine designing phase, the shortlisted epitopes were connected through GPGPG linkers and the generated peptide was linked to cholera toxin B subunit adjuvant (CTBS) by another “EAAAK” linker. Linkers allow efficient separation of the epitopes. The designed vaccine construct was subjected to physicochemical properties analysis. The server predicted that the designed vaccine construct comprises 211 amino acids. The molecular weight of the molecule is 22.64 kDa, theoretical PI value is 9.27, and instability index is 39.85 (stable). Furthermore, aliphatic index of the vaccine is 70.05 and grand average of hydropathicity (GRAVY) is −0.428. The control ompA molecule has a molecular weight of 37.2 kDa, a theoretical pI value of 5.99, an instability index of 21.44, and a GRAVY score of −0339. All these values indicate ompA as potential vaccine target. The results of ompA are similar to that of the vaccine molecule designed in this study; therefore, we can predict the vaccine is a potential vaccine candidate for experimental evaluation. 

### 3.8. Structure Prediction and Loops Refinement 

The 3D structure of the vaccine was predicted using sequences of the multi-epitope-vaccine construct. The vaccine construct comprises cholera toxin B subunit adjuvant, EAAK, and GPGPG linkers and selected epitopes. The 3D structure is presented in Figure 5 while the schematic representation is shown in Figure 6. Furthermore, the loops present in the vaccine structure were further refined in order to maintain the structure’s stability. The galaxyWeb webserver generated 10 refine models based on RMSD, MolProbity, clash score, poor rotamers, Rama favored residues percentage, and GALAXY energy (Table 4)

### 3.9. Disulfide Engineering and In-Silico Codon Optimization 

Disulfide engineering analysis reported 16 amino acid residues that could be replaced by cysteine amino acid. The mutated pair of amino acids are represented by yellow colored stick in the vaccine structure (Figure 7) and tabulated in Table 5. Next, the codon optimization was performed where the reverse translated DNA sequence “ATGATCAAACTGAAATTTGGCGTCTTCTTCACCGTCCTGCTGTCTTCTGC TTACGCTCACGGTACCCCGCAGAACATCACCGACCTGTGCGCTGAATACC ACAACACCCAGATCTACACCCTGACAAAATCTTCTCTTACAGAATCTCTGGCTGGTAAACGTGAAATGGCTATCATCACCTTCAAAAACGGTGCTATCTTCCAGGTTGAAGTTCCGGGTTCTCAGCACATCGACTCTCAGAAAAAAGCTATCGAAGTATGAAAGACACCCTGCGTATCGCTTACCTGACGAGCTAAAGTGAAAAACTGTGCGTGAACAACAAAACCCCGCACGCTATCGCTGCTATCTCTATGGCTAACGAGCTGCTGCTGAAGTATGCCGAAGTATGCCGGCTGCTGGTCCGGGTCCGGGTCGTTCTTCTCCGCCGGCTGCTGGTGCTGGTCCGGGTCCGGGTGACAAGTCCGATCTTATCAACTGGGTCCGGGTCCGGGTCGTCAGCGTTTCGAGCTCAGCTCGTGGTCCGGGTCCGGGTGCTGAAGTGAGCGTCAGAGGTTCGACGCTGGTCCAGGTCCGGGTCACGCTCGTGCTGCTCAGCTGGAACCGCTG” was inserted into the pET28a(+) vector. As shown in Figure 8, the DNA sequences are represented by magenta color. The antigenicity score of disulfide-engineered vaccine is 0.6952, indicating a good overall antigenicity of the sequence. 

### 3.10. Secondary Structure Prediction, Z-Score Calculation and Ramachandran Plot Analysis 

The secondary structure was predicted using the PDBsum generate tool as shown in Figure 9A. Secondary structure of the multi-epitope-vaccine construct revealed that 84 (39.8%) of the residues are alpha helix, 5 residues have 3–10 helixes (2.4%), and 122 have other helixes (57.8%). The multi-epitope-3D statistics describe that most of the vaccine residues are in favored regions. A total of 10 residues were in additional allowed regions (7.0%), 2 were in generously allowed regions (1.4%), 1 was in a disallowed regions (0.7%), and 143 were non-glycine and non-proline residues (Figure 9B). The Z-Score of the vaccine is −1.65, as shown in Figure 9C. 

### 3.11. Agreescan3D and CABS-Flex 2.0 Analysis 

The vaccine candidate has a minimal score value of −4.71, a maximal score value of 3.14, an average score of −0.80, and a total score value of −169.32. The Aggrescan3D superimposed structures are shown in Figure 10A. The vaccine candidate was further found in 10 models that were generated using simulation. The vaccine candidate was found to show a maximum RMSF of 6.22 Å and a minimum RMSF of 0.9 Å. The vaccine candidate RMSF plot is presented in Figure 10B.

### 3.12. Binding Interaction Analysis 

A docking approach was utilized to check vaccine binding and interactions with the immune cell receptors MHC-I, MHC-II, and TLR-4 chosen as the selected immune cell receptors, which play important role in antigen presentation and processing. In each case, the server generated 10 docked complexes based on the binding energy score as mentioned in Table 6, Table 7 and Table 8. Moreover, intermolecular docked complexes are provided Figure 11A–C. In the case of the vaccine with MHC-I, the selected complex has a lowest energy of −944.1 kcal/mol; for the vaccine with MHC-II, the selected complex has a lowest energy of −933.1 kcal/mol; and in the case of the vaccine with TLR-4, the selected docked complex has a lowest energy of −1067.3 kcal/mol. These complexes were considered best for simulation. For MHC-I, the vaccine docked at the active pocket. For MHC-II, the vaccine interacts near the active pocket region and the important epitopes are exposed.

### 3.13. Molecular Dynamic Simulation Analysis 

Molecular dynamic analysis is a computer-based simulation for assessing the dynamic movement of docked molecules. The molecules and atoms are simulated for a given period of time and the dynamics are investigated using variety of statistics tests. Newton’s equation of motion is applied to determine movement of atoms and molecules. In this analysis, the docked complexes (vaccine-receptors) were analyzed for 100 nanoseconds. In simulation time, the important steps are to evaluate the binding efficacy and stability mode of the docked molecules. The simulation analysis of vaccine-receptors complexes is given in Figure 12. The first analysis which was performed in the simulation was RMSD, which was performed based on carbon’s alpha atoms. In the RMSD analysis, it was observed that the vaccine and TLR-4 docked complex showed the best binding affinity followed by the vaccine-MHC-I and the vaccine-MHC-II, as shown in Figure 12A. The mean RMSD of the vaccine with TLR-4 was 3.5 angstrom, while for the vaccines with MHC-I and MHC-II, the average RMSD was 4.5 angstrom and 5.1 angstrom, respectively. Little deviations were seen in the systems due to the large size and the presence of loops in the structures. Following RMSD, RMSF analysis was performed in order to analyze residue level fluctuations. The RMSF plot is given in Figure 12B. The majority of the residues were in the stable range; however, the vaccine with TLR-4 showed some high deviations. These deviations are due to the vaccine attempting to acquire more stable conformation with the receptor. Nevertheless, the vaccine remained in constant contact with TLR-4 throughout the entire simulation. 

### 3.14. Binding Free Energy Calculation

The binding interactions of docked complexes were also analyzed using the MM-GBSA method for the binding free energies calculation. In MM-GBSA analysis, different energy parameters were calculated. The estimated net binding free is −23.98 kcal/mol, −16.84 kcal/mol, and −15.50 kcal/mol for vaccine-TLR-4, vaccine-MHC-I, and vaccine-MHC-II, respectively. The different energies are mentioned in Table 9.

## 4. Discussion

*B. mallei* is the etiological agent of Melioidosis disease, which is also known as Whitmore’s disease [41]. Reports have been documented that suggest the speedy evolution of antibiotic resistance mechanisms and, due to non-availability of approved vaccine against the pathogen, serious efforts are needed to develop novel therapeutic strategies. Development of a multi-epitope vaccine is a promising approach as it may prevent the pathogen’s spread and overcome its infections. In the current research work, a multi-epitope-based vaccine was constructed against *B. mallei* by using RV and immunoinformatics approaches [42]. A previous in silico study conducted by Saba et al. designed a multi-epitope-based vaccine against *Providencia rettgeri* that showed promising potency in terms of generating proper immune responses against the targeted pathogen (doi: 10.3390/vaccines10020189). In the present study, the complete proteome of the pathogen was utilized for identification of good vaccine candidates [28]. Complete proteomes were utilized in order to make a potent broad-spectrum vaccine candidate against all available sequenced strains. Core proteins are present among all the strains, so we retrieved the core sequence and processed it for surface localized proteins. Surface localized proteins are exposed to the immune system and can evoke proper immune responses as they contain antigenic determinants [43]; therefore, only surface localized outer membrane, extracellular membrane, and periplasmic membrane proteins were considered to be good vaccine candidates. Virulent proteins are mainly involved in the pathogenicity of pathogens and simulate effective immune reactions, so virulent proteins were filtered [27]. Multi-epitope vaccines consist of different B and T-cell epitopes in order to generate both humoral and cellular immune responses in the host body against a pathogen. Epitope prediction and prioritization were completed for screening and targeting of probable antigenic epitopes. To increase the antigenicity of the proposed vaccine construct, cholera toxin B (CTB) was used as an adjuvant and linked to epitope peptides at the N-terminus via the EAAAK linker. CTB is a non-toxic component of the cholera toxin that attaches to dendritic cells, B cells, and macrophages. Its optimal immune system access is made possible by its affinity for the monosialotetrahexosylganglioside (GM1), which is found in a wide range of cell types including gut epithelial cells, antigen-presenting cells, macrophages, dendritic cells, and B cells. Many different organisms can easily express CTB on its own. Different methods can be employed to link this adjuvant to antigens either through genetic fusion or chemical manipulation, leading to much improved immune responses to the antigens (doi: 10.3390/vaccines3030579). The 3D structure modeling and validation is important, so the 3D structure was modeled. To retain the structure stability, the structure was further refined because structure stability of vaccine candidate is important. The multi-epitope vaccine showed good physicochemical features in terms of thermodynamic feasibility, stability, hydrophilicity, and expression capacity. The multi-epitope vaccine is non-allergen; thus, harmful responses of the vaccine are not expected. The vaccine designed in this study exhibited a high level of antigenicity, which is much preferred for immunological applications. In addition, overexpression of this vaccine could be undertaken in *Escherichia coli* K12 strain. To generate immune responses against the vaccine antigen, the vaccine should interact with host immune cells. Hence, we conducted a docking study in order to validate the docking potency of vaccine candidates with MHC-I, MHC-II, or TLR-4. The same study conducted by Ismail et al. designed of a multi-epitope-based vaccine against nosocomial Enterobacteriaceae pathogens by applying pan-genome based RV method [27]. The findings of this study are new and may speed up vaccine designs against *B. mallei*. This could save money, save time, and save human efforts. Therefore, it is strongly suggested to use the designed vaccine construct in in vivo and in vitro studies and disclose its real immune protective capacity. 

## 5. Conclusions

As concluding remarks, this study has proven the antigenicity of one extracellular (type VI secretion system tube protein (Hcp)) and one outer membrane (type IV pilus secretin (PilQ)). The proteins were then subjected to shortlist epitopes for designing a multi-epitope vaccine construct against *B. mallei*. The complete proteomes were scanned to identify immunodominant epitopes that can induce both humoral and cellular immune response against the pathogen. By employing several immunoinformatics tools, several epitopes were shortlisted for vaccine construction. The designed vaccine construct showed stable physicochemical, antigenic, good water soluble, and non-allergenic properties. The vaccine construct comprises immunogenic and putatively harmless and safe epitopes for prophylactics and therapeutic vaccine formulations. The modeled 3D structure of the designed vaccine constructs further confirmed that the structure is stable. Moreover, the designed vaccine successfully binds to the selected immune cells receptors (MHC-I, MHC-II, and TLR-4); therefore, it proficiently triggers both the cellular and humoral immune responses against targeted pathogen. It was also observed that the vaccine formed strong van der Waals and electrostatic chemical interactions with immune receptors, and thus formed stable complexes, which further increased vaccine epitope presentation and immune response generation. The designed vaccine construct still requires experimental analysis in order to confirm its potency against *B. mallei* infections.

## Figures and Tables

**Figure 1 vaccines-10-01580-f001:**
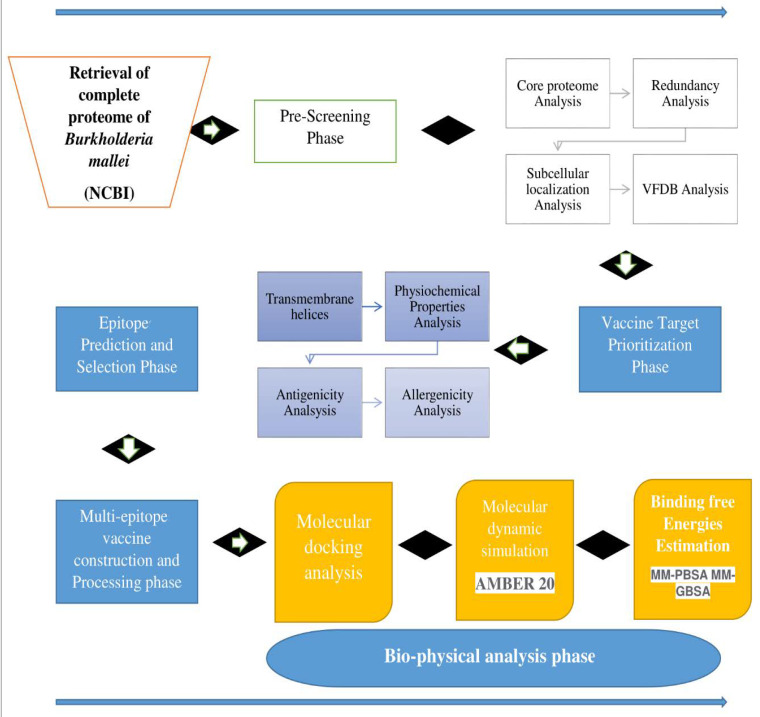
Schematic representation of research methodology followed for vaccine construction against *B. mallei*.

**Figure 2 vaccines-10-01580-f002:**
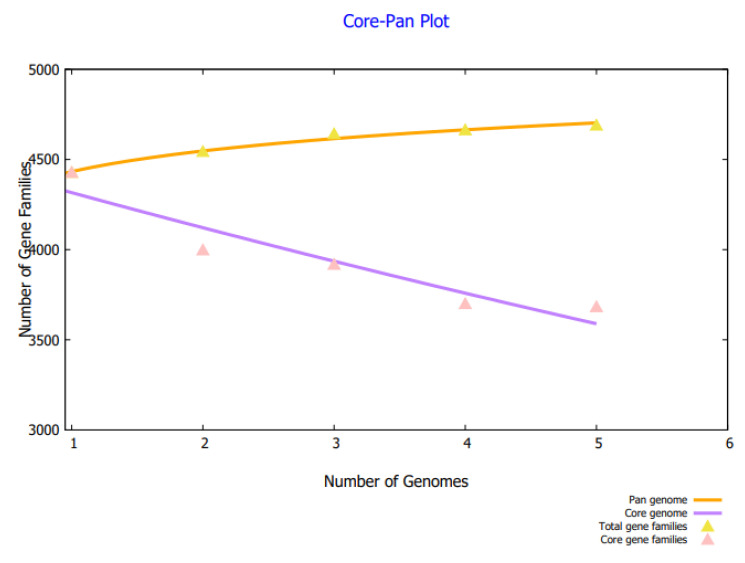
Core-pan plot illustrating the number of gene families in each genome. The pan and core gene families can be differentiated by different colors.

**Figure 3 vaccines-10-01580-f003:**
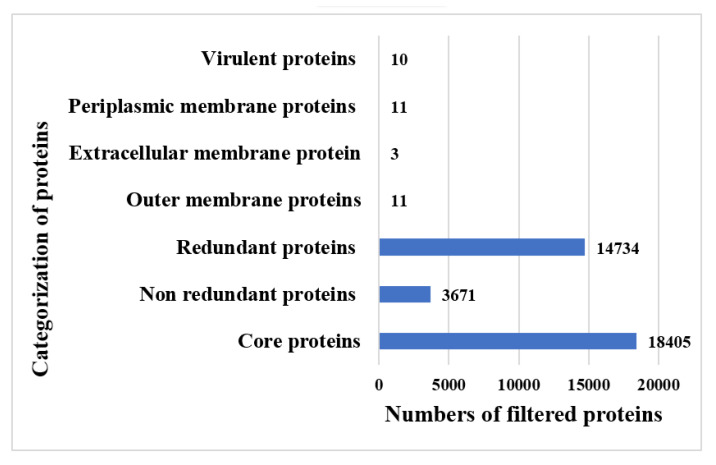
Numbers and categories of proteins filtered in each phase of subtractive proteomics.

**Figure 4 vaccines-10-01580-f004:**
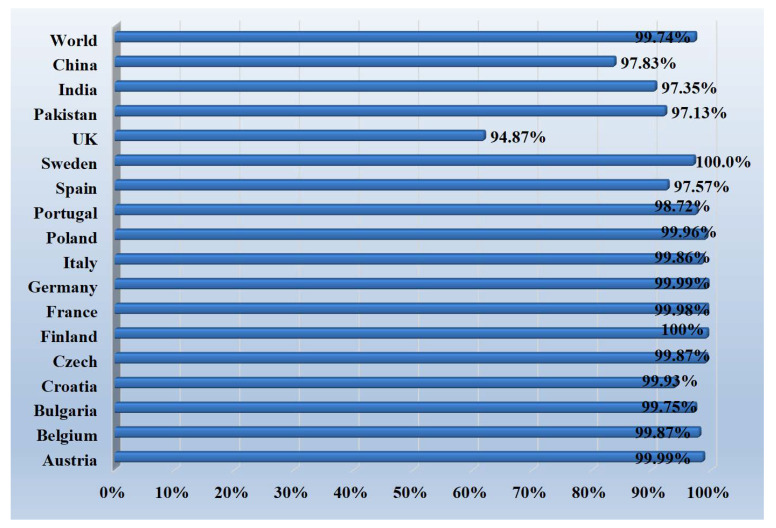
Population coverage analysis of selected vaccine epitopes.

**Figure 5 vaccines-10-01580-f005:**
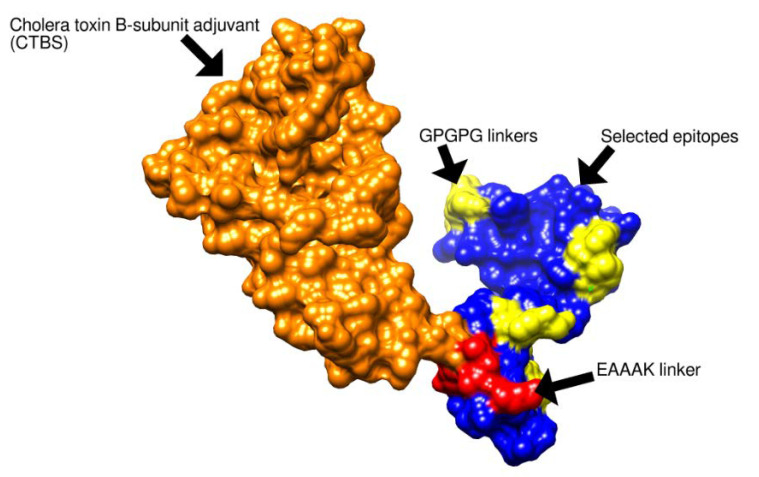
The 3D structure of multi-epitopes vaccine. Each segment is properly labeled in the figure.

**Figure 6 vaccines-10-01580-f006:**
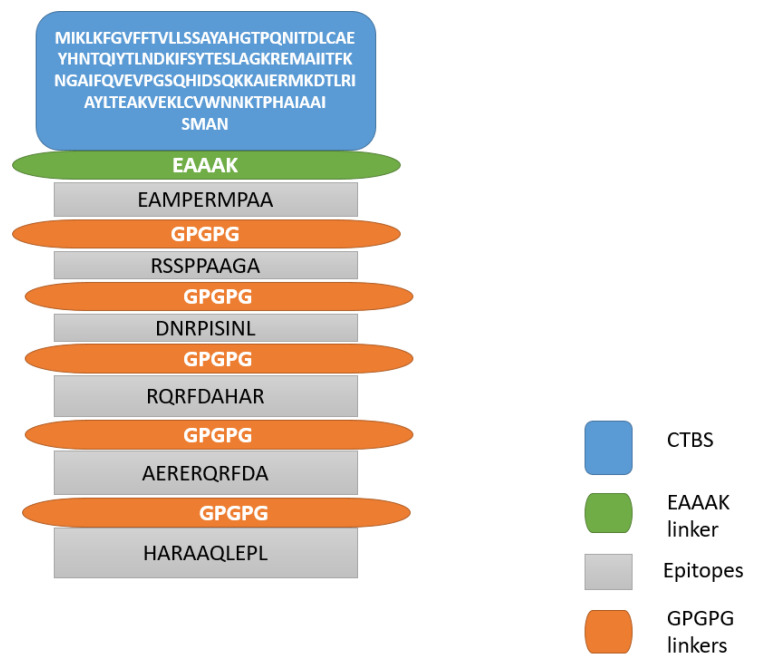
Schematic representation of multi-epitopes vaccine.

**Figure 7 vaccines-10-01580-f007:**
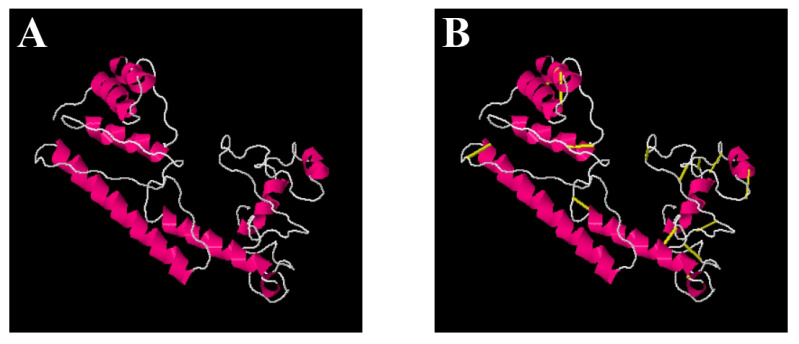
(**A**) Original structure of vaccine molecule. (**B**) Mutated structure.

**Figure 8 vaccines-10-01580-f008:**
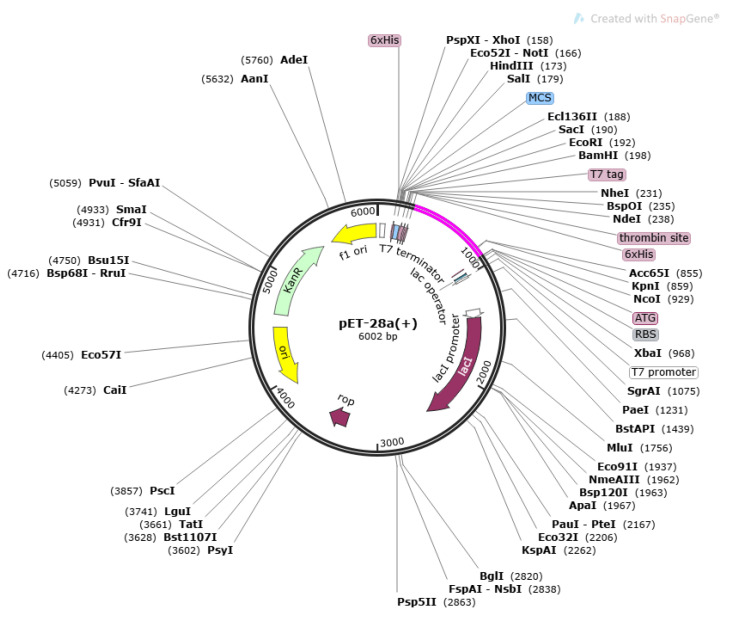
Cloned pET-28a(+) vector. The magenta color represents inserted DNA sequence.

**Figure 9 vaccines-10-01580-f009:**
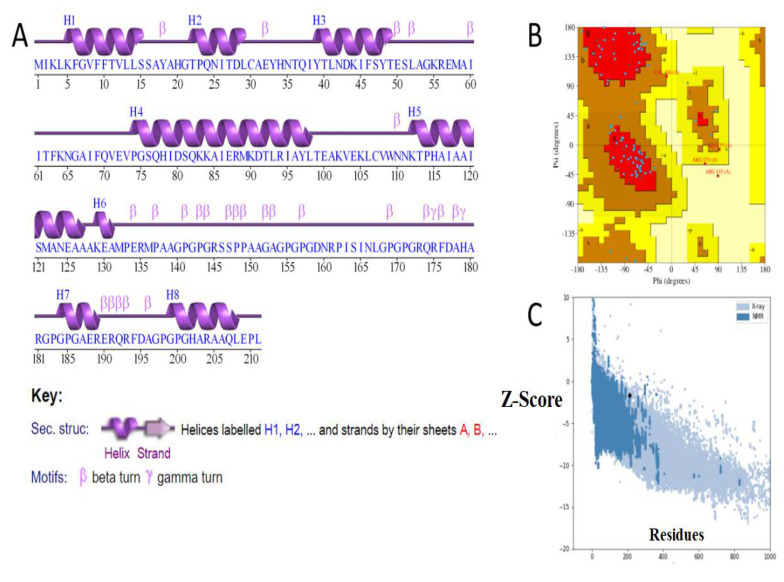
Structural analysis of the vaccine construct. (**A**) Secondary structure; (**B**) Ramachandran plot; and (**C**) Z-score plot.

**Figure 10 vaccines-10-01580-f010:**
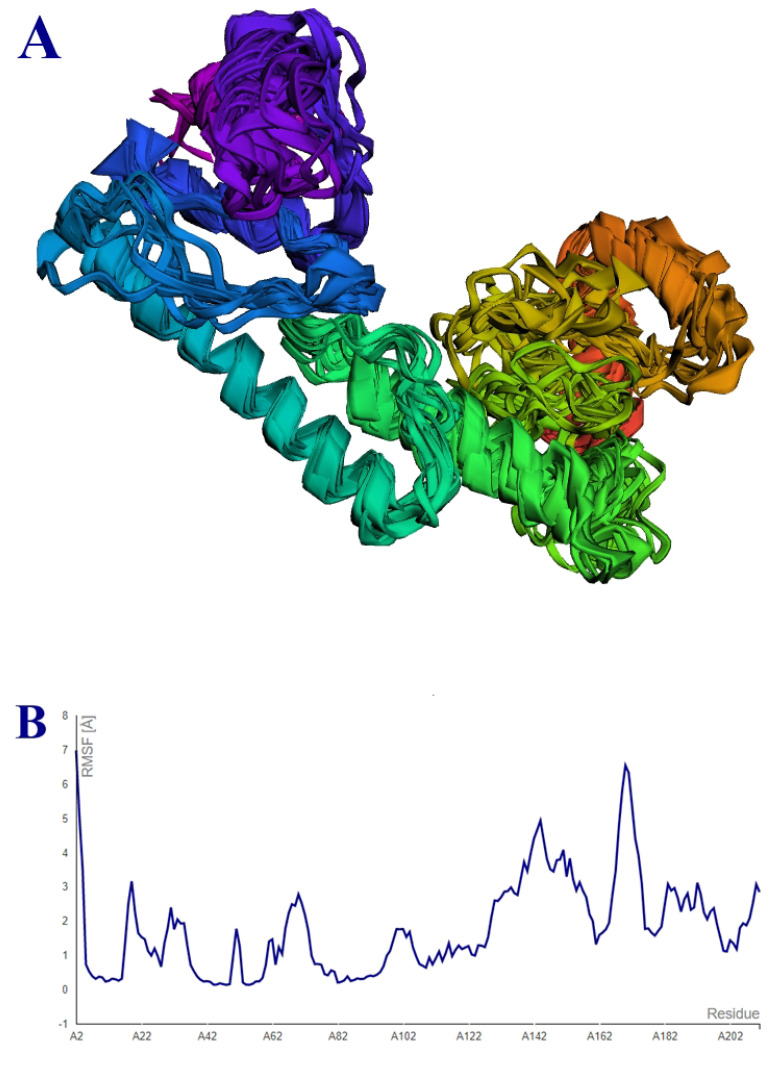
(**A**) Superimposed 3D structure. (**B**) RMSF graph.

**Figure 11 vaccines-10-01580-f011:**
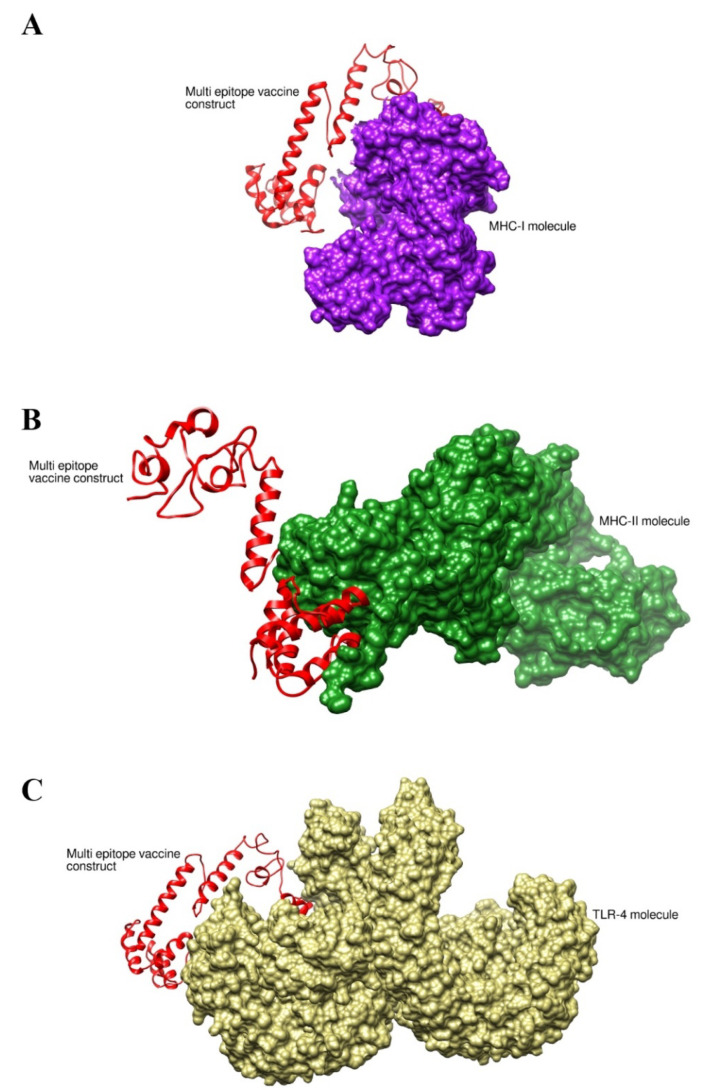
Binding mode of vaccine with different immune receptors. (**A**) Vaccine with MHC-I; (**B**) vaccine with MHC-II; and (**C**) vaccine with TLR-4.

**Figure 12 vaccines-10-01580-f012:**
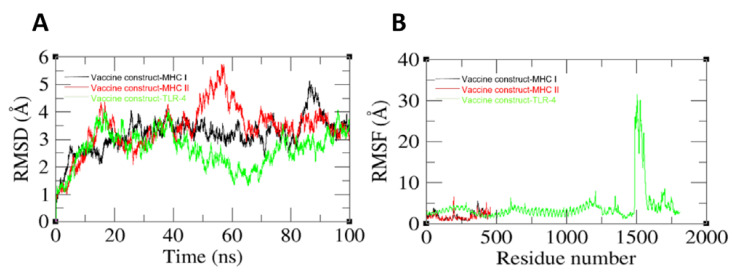
Simulation trajectories analysis to evaluate vaccine stability with the immune receptors. (**A**) RMSD. (**B**) RMSF.

**Table 1 vaccines-10-01580-t001:** Predicted B-cells epitopes.

Target Proteins	Predicted B-Cells Epitopes
core/3507/1/Org1_Gene145(type VI secretion system tube protein Hcp	ASQPGAMASGSGGNAGKASF
	KQYWQQNDNGGKGAEVSVGWNIKE
core/426/1/Org1_Gene4503 (type IV pilus secretin PilQ protein)	EAVASLPPLPVGAPFGWSASASVGAAGRAPLPEAAAPQWRFDSARDPVAGAPSPDVDGGAPAAEFAGEAMPERMP AAPTAEPARSTSADAGTSSAVASAGLQAQ EAALEGPPVPLAPAQRMSDESDEHRSSPPAAGAVSTASVAGTGTETGDPSGDNRPISINLQQAS
	VAELAERERQRFDAHARAAQLEPLASRG
	LAGSAGQRILSKRGSVLA
	RGFSRNLGARLALRAPDAGERATGIVAGRNGTLAELAARPISGFDAATAGLTLFAARASRL
	SDDRDDVTRVPLL

**Table 2 vaccines-10-01580-t002:** T-cells epitopes with lowest percentile score.

Major Histocompatibility Complex II (MHC-II)	Percentile Score	Major Histocompatibility Complex I(MHC-I)	Percentile Score
ASQPGAMASGSGGN	6	ASQPGAMAS	3.5
AMASGSGGNAGKASF	8	ASGSGGNAGK	0.7
GGKGAEVSVGWNIK	26	KGAEVSVGWN	2.8
QYWQQNDNGGKGAEV	34	DNGGKGAEV	4.2
KQYWQQNDNGGKGAE	32	KQYWQQNDN	9.3
PVGAPFGWSASASVGA	18	APFGWSASA	0.25
GAAGRAPLPEAAAPQWR	13	LPEAAAPQW	0.01
FDSARDPVAGAPSPDVDGG	10	DSARDPVAGA	0.29
DGGAPAAEFAGEAMPERMPAA	0.58	AMPERMPAA	0.06
DAGTSSAVASAGLQAQEAALE	8.16	GLQAQEAAL	0.48
GPPVPLAPAQRMSDESDE	23	VPLAPAQRM	0.01
ESDEHRSSPPAAGAVSTAS	8.56	RSSPPAAGA	0.33
SGDNRPISINLQQAS	5.4	DNRPISINL	0.49
AERERQRFDAHARA	15	RQRFDAHAR	0.21
HARAAQLEPLASRG	22	AQLEPLASR	0.6
AGQRILSKRGSVLA	2.7	ILSKRGSVL	0.7
RGFSRNLGARLALR	0.01	RGFSRNLGAR	0.4
APDAGERATGIVAGRNG	79	ERATGIVAGR	0.6
TLAELAARPISGFD	20	LAARPISGF	0.49
AGLTLFAARASRL	0.68	TLFAARASR	0.01
SDDRDDVTRVPLL	19	DDVTRVPLL	1.3

**Table 3 vaccines-10-01580-t003:** Selected proteins and their good vaccine candidate properties.

Selected Epitopes	Predicted IC50 Value (nM) Score	Antigenicity	Allergenicity	Water Solubility	Toxicity
EAMPERMPAA	6.28	0.7304	Non-allergen	Good water soluble	Non-toxin
RSSPPAAGA	6.41	0.8995			
DNRPISINL	17.38	1.1305			
RQRFDAHAR	9.27	0.8286			
AERERQRFDA	23.55	0.8414			
HARAAQLEPL	4.72	1.1458			

**Table 4 vaccines-10-01580-t004:** Top refine models generated by GalaxyWeb webserver.

Model	RMSD	Mol Probity	Clash Score	Poor Rotamers	Rama Favored	GALAXY Energy
Initial	0.000	3.689	124.8	6.6	91.4	28,723.56
MODEL 1	3.679	1.487	2.9	0.0	93.8	−3390.36
MODEL 2	3.058	1.594	3.7	0.0	93.3	−3373.64
MODEL 3	2.887	1.548	2.6	0.0	91.4	−3361.08
MODEL 4	0.992	1.654	3.4	0.0	90.9	−3353.07
MODEL 5	3.741	1.691	4.0	0.6	91.4	−3352.74
MODEL 6	1.194	1.521	3.7	0.0	94.7	−3348.47
MODEL 7	2.506	1.642	4.3	0.0	93.3	−3344.80
MODEL 8	0.939	1.406	2.9	0.0	95.2	−3343.12
MODEL 9	0.929	1.466	3.1	0.0	94.7	−3341.38
MODEL 10	3.104	1.494	3.4	0.0	94.7	−3339.60

**Table 5 vaccines-10-01580-t005:** Amino acid residues that are replaced by cysteine amino acid.

Pair of Amino Acid Residues	Chi3 Value	Energy
PHE9-ALA31	−65.14	5.86
SER16-THR27	98.73	1.12
ILE38-LEU41	87.1	2.09
VAL71-GLY75	125.46	5.24
TRP109-LYS112	114.42	3.8
ALA123-ALA153	72.87	3.43
GLU125-ALA131	115.02	3.97
ALA128-ALA131	111.84	2.53
PRO143-PRO149	−66.45	4.67
PRO155-ASN160	94.44	3.65
ILE165-ALA196	123.98	8.9
GLY168-PRO171	−114.69	4.22
ALA178-ALA187	117.98	7.08
ALA178-ARG191	−93.98	0.38
GLY182-GLY186	103.77	0.3
ASP195-PRO198	99.88	1.93

**Table 6 vaccines-10-01580-t006:** Docking score of vaccine-MHC-I solutions.

Cluster	Members	Representative	Weighted Score
0	51	Center	−773.1
		Lowest Energy	−944.1
1	47	Center	−760.8
		Lowest Energy	−824.1
2	44	Center	−783.9
		Lowest Energy	−798.3
3	35	Center	−753.1
		Lowest Energy	−890.3
4	34	Center	−760.4
		Lowest Energy	−904.0
5	32	Center	−752.1
		Lowest Energy	−952.1
6	32	Center	−833.9
		Lowest Energy	−1027.7
7	30	Center	−841.1
		Lowest Energy	−841.1
8	28	Center	−725.5
		Lowest Energy	−860.2
9	26	Center	−862.9
		Lowest Energy	−942.2
10	25	Center	−722.8
		Lowest Energy	−933.1

**Table 7 vaccines-10-01580-t007:** Docking score of vaccine-MHC-II solutions.

Cluster	Members	Representative	Weighted Score
0	99	Center	−895.4
		Lowest Energy	−975.5
1	79	Center	−920.5
		Lowest Energy	−1108.2
2	71	Center	−938.3
		Lowest Energy	−1076.4
3	62	Center	−937.8
		Lowest Energy	−1232.3
4	34	Center	−990.6
		Lowest Energy	−990.6
5	30	Center	−929.9
		Lowest Energy	−1043.2
6	25	Center	−984.5
		Lowest Energy	−989.1
7	22	Center	−837.2
		Lowest Energy	−940.4
8	18	Center	−838.7
		Lowest Energy	−953.3
9	17	Center	−995.2
		Lowest Energy	−995.2
10	17	Center	−837.4
		Lowest Energy	−985.0

**Table 8 vaccines-10-01580-t008:** Docking score of vaccine-TLR-4 solutions.

Cluster	Members	Representative	Weighted Score
0	89	Center	−859.1
		Lowest Energy	−1067.3
1	50	Center	−888.9
		Lowest Energy	−1002.0
2	49	Center	−845.6
		Lowest Energy	−946.1
3	45	Center	−859.9
		Lowest Energy	−1003.6
4	35	Center	−915.8
		Lowest Energy	−1021.9
5	28	Center	−871.6
		Lowest Energy	−941.0
6	26	Center	−920.1
		Lowest Energy	−966.6
7	26	Center	−853.4
		Lowest Energy	−1032.2
8	24	Center	−838.6
		Lowest Energy	−974.0
9	24	Center	−906.1
		Lowest Energy	−999.5
10	23	Center	−868.2
		Lowest Energy	−971.4

**Table 9 vaccines-10-01580-t009:** MM-GBSA binding energy calculation. The energy values are described in kcal/mol.

Energy Parameter	TLR-4-Vaccine Complex	MHC-I-Vaccine Complex	MHC-II-Vaccine Complex
MM-GBSA			
VDWAALS	−33.5184	−26.2334	−22.3071
EEL	−153.63	−12.0301	−225.981
EGB	167.4479	24.8686	235.4421
ESURF	−4.2856	−3.4476	−2.6587
Delta G gas	−187.149	−38.2635	−248.288
Delta G solv	163.1624	21.421	232.7834
Delta Total	−23.9861	−16.8425	−15.5046

## Data Availability

Not applicable.

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
