# Peer review of "Computational Based Designing of a Multi-Epitopes Vaccine against Burkholderia mallei"

_vaccines, 2022, doi:10.3390/vaccines10101580_

Round 1
Reviewer 1 Report
This manuscript described a multi-epitopes vaccine against Burkholderia mallei that was designed and validated using several bioinformatics tools. Six B/T cell epitopes were selected and combined in the vaccine design, which was then shown to bind immune cell receptors based on simulation tools. However, this paper has multiple critical flaws.
· The entire manuscript is based on predictions from bioinformatic tools, but wet lab data is lacking to validate the accuracy of these predictions, making the overall conclusions extremely unconvincing.
· How conserve are the predicted epitopes in different strains of B. mallei?
· Some of the cystine mutations are located in the cholera toxin B subunit, and some are in the middle of the predicted epitopes, would these mutations compromise the efficacy of the vaccine?
· MHC molecules are polymorphic, and the binding specificity is dictated by HLA genes. I’m not sure how the authors can predict the interactions with random MHC-I/II molecules. Are the interactions between the antigen and MHC molecules docked at the MHC binding pockets? In addition, the predicted lowest energy/docking scores are meaningless without a negative control, I’m not convinced that these interactions are real.
Author Response
Comments and Suggestions for Authors
This manuscript described a multi-epitopes vaccine against Burkholderia mallei that was designed and validated using several bioinformatics tools. Six B/T cell epitopes were selected and combined in the vaccine design, which was then shown to bind immune cell receptors based on simulation tools. However, this paper has multiple critical flaws.
The entire manuscript is based on predictions from bioinformatic tools, but wet lab data is lacking to validate the accuracy of these predictions, making the overall conclusions extremely unconvincing.
Response: We understand the reviewer concern. As the scope of the paper is computational, the main objective of the work is to provide theoretical vaccine model for experimentalists to check the designed vaccine immune protective efficacy against the said pathogen in vivo. Reverse Vaccinology (RV), has received more attention in recent years and has been used for the identification of vaccine proteins against different pathogens [1]. The RV approach was first applied to the bacterial pathogen Meningococcus B (MenB) and led to the license Bexsero vaccine [2], where RV played a significant role in screening for an antigen with the broadest bactericidal activity and ultimately resolved the long journey of MenB vaccine development. RV has also been applied to many other bacterial pathogens, including group A Streptococcus, antibiotic-resistant Staphylococcus aureus, Streptococcus pneumonia, and Chlamydia. The efficacy of peptide or subunit-based vaccines initially identified through a RV protocol has also been proven experimental [3,4]. In this study, a RV approach was used to screen possible vaccine proteins against the Burkholderia mallei identifying extracellular proteins as strong candidates for vaccine development. Experimental follow up by testing the immune protection efficacy of the screened epitopes in animal models will open for experimentalists and this study will definitely speed up vaccine development process against this pathogen. This text has been highlighted in the conclusion section of the revised manuscript.
How conserve are the predicted epitopes in different strains of B. mallei?
Response: We Understand the reviewer’s concern. In this study, we retrieve all completely sequenced available strains and subjected them to Bacterial Pan Genome Analysis(BPGA) to extract the core proteome of the Burkholderia mallei. The predicted epitopes are conserved in all available strains B. mallei. Please refer to section 2.1 and 3.2 of the manuscript for more details.
Some of the cystine mutations are located in the cholera toxin B subunit, and some are in the middle of the predicted epitopes, would these mutations compromise the efficacy of the vaccine?
Response:
Thank you for the reviewer’s response and concern. In order to check the effect of disulfide engineering on the antigenicity of the MEV construct, antigenicity of the mutant vaccine is evaluated via VaxiJen 2.0 web-tool. The antigenicity of the mutant construct is 0.6758 which is nearly similar to that of original construct whose antigenicity is 0.6952. This proves that disulfide engineering will enhance the stability of the MEV without having a drastic effect on antigenicity. Please refer to section 3.9 of the manuscript.
MHC molecules are polymorphic, and the binding specificity is dictated by HLA genes. I’m not sure how the authors can predict the interactions with random MHC-I/II molecules. Are the interactions between the antigen and MHC molecules docked at the MHC binding pockets? In addition, the predicted lowest energy/docking scores are meaningless without a negative control, I’m not convinced that these interactions are real.
Response: Thank you for this good comment. The section 2.2 is revised as per reviewer comment. The prediction of T cell antigens is based on reference set of MHC-I and MHC-II molecules provided at IEDB server. Please refer to the links http://tools.iedb.org/mhci/ for MHC-I and http://tools.iedb.org/mhcii/ for MHC-II. Secondly the docking presents the intermolecular interactions between the vaccine and MHC molecules. For MHC-I, the vaccine docked at the active pocket while for MHC-II, the vaccine interacts near active pocket region and important the epitopes are exposed. About the interactions these are predictions which need cross validation by experimental studies but these results will speed up the experimental work on the designed vaccine molecule.
Reviewer 2 Report
The authors bring a very interesting topic regarding the development of vaccine against Burkholderia mallei. However, the manuscript does not fit with the scope from the journal. My sugestion to the authors is to submit the manuscript to a journal more related with bioinformatic approches.
Author Response
Reviewer # 2
Comments and Suggestions for Authors
The authors bring a very interesting topic regarding the development of vaccine against Burkholderia mallei. However, the manuscript does not fit with the scope from the journal. My sugestion to the authors is to submit the manuscript to a journal more related with bioinformatic approaches.
Response: We understand the reviewer concern. As the scope of the paper is computational, the main objective of the work is to provide theoretical vaccine model for experimentalists to check the designed vaccine immune protective efficacy against the said pathogen in vivo. Reverse Vaccinology (RV), has received more attention in recent years and has been used for the identification of vaccine proteins against different pathogens [1]. The RV approach was first applied to the bacterial pathogen Meningococcus B (MenB) and led to the license Bexsero vaccine [2], where RV played a significant role in screening for an antigen with the broadest bactericidal activity and ultimately resolved the long journey of MenB vaccine development. RV has also been applied to many other bacterial pathogens, including group A Streptococcus, antibiotic-resistant Staphylococcus aureus, Streptococcus pneumonia, and Chlamydia. The efficacy of peptide or subunit-based vaccines initially identified through a RV protocol has also been proven experimental [3,4]. In this study, a RV approach was used to screen possible vaccine proteins against the Burkholderia mallei identifying extracellular proteins as strong candidates for vaccine development. Experimental follow up by testing the immune protection efficacy of the screened epitopes in animal models will open for experimentalists and this study will definitely speed up vaccine development process against this pathogen. This text has been highlighted in the conclusion section of the revised manuscript.
Many such works of ours are published in this Vaccines journal and the paper is in lines of the journal scope. Please refer to the following articles recently published in the said journal having similar Immune-informatics approach.
- Yousaf, M., Ullah, A., Sarosh, N., Abbasi, S. W., Ismail, S., Bibi, S., ... & Bin Emran, T. (2022). Design of Multi-Epitope Vaccine for Staphylococcus saprophyticus: Pan-Genome and Reverse Vaccinology Approach. Vaccines, 10(8), 1192.
- Alshabrmi, F. M., Alrumaihi, F., Alrasheedi, S. F., Al-Megrin, W. A. I., Almatroudi, A., & Allemailem, K. S. (2022). An In-Silico Investigation to Design a Multi-Epitopes Vaccine against Multi-Drug Resistant Hafnia alvei. Vaccines, 10(7), 1127.
- Albekairi, T. H., Alshammari, A., Alharbi, M., Alshammary, A. F., Tahir ul Qamar, M., Anwar, T., ... & Ahmad, S. (2022). Design of a Multi-Epitope Vaccine against Tropheryma whipplei Using Immunoinformatics and Molecular Dynamics Simulation Techniques. Vaccines, 10(5), 691.
- Ismail, S., Abbasi, S. W., Yousaf, M., Ahmad, S., Muhammad, K., & Waheed, Y. (2022). Design of a Multi-Epitopes Vaccine against Hantaviruses: An Immunoinformatics and Molecular Modelling Approach. Vaccines, 10(3), 378
References
- Ong E, Wong MU, Huffman A, He Y. COVID-19 coronavirus vaccine design using reverse vaccinology and machine learning. bioRxiv [Preprint]. 2020 Mar 21:2020.03.20.000141. doi: 10.1101/2020.03.20.000141. Update in: Front Immunol. 2020 Jul 03;11:1581. PMID: 32511333; PMCID: PMC7239068.
- Folaranmi T, Rubin L, Martin SW, Patel M, MacNeil JR; Centers for Disease Control (CDC). Use of Serogroup B Meningococcal Vaccines in Persons Aged ≥10 Years at Increased Risk for Serogroup B Meningococcal Disease: Recommendations of the Advisory Committee on Immunization Practices, 2015. MMWR Morb Mortal Wkly Rep. 2015 Jun 12;64(22):608-12. Erratum in: MMWR Morb Mortal Wkly Rep. 2015 Jul 31;64(29):806. PMID: 26068564; PMCID: PMC4584923.
- Maione D, Margarit I, Rinaudo CD, Masignani V, Mora M, Scarselli M, Tettelin H, Brettoni C, Iacobini ET, Rosini R, D'Agostino N, Miorin L, Buccato S, Mariani M, Galli G, Nogarotto R, Nardi-Dei V, Vegni F, Fraser C, Mancuso G, Teti G, Madoff LC, Paoletti LC, Rappuoli R, Kasper DL, Telford JL, Grandi G. Identification of a universal Group B streptococcus vaccine by multiple genome screen. Science. 2005 Jul 1;309(5731):148-50. doi: 10.1126/science.1109869. Erratum in: Science. 2013 Jan 11;339(6116):141. Nardi Dei, Vincenzo [corrected to Nardi-Dei, Vincenzo]. PMID: 15994562; PMCID: PMC1351092.
- Sette A, Rappuoli R. Reverse vaccinology: developing vaccines in the era of genomics. Immunity. 2010 Oct 29;33(4):530-41. doi: 10.1016/j.immuni.2010.09.017. PMID: 21029963; PMCID: PMC3320742.
Reviewer 3 Report
This is very interesting work and I have been privileged to review. few things need to considered:
- All organism names should be italic all all across the entire document
- Fig, 1all texts inside boxes should be in vertical direction and relatedness between the last three boxes (yellow) should be clarified by arrows direction like others.
- Table 2. title need to be moved to the next page
- Section 3.6 and Fig 4 need to be clarified. Why certain populations and countries are presented over others?
- Table 7, line 305 to be moved to the next page
Good luck
Author Response
Comments and Suggestions for Authors
This is very interesting work and I have been privileged to review. few things need to considered:
- All organism names should be italic all all across the entire document
Response: Corrected in revised manuscript
- Fig, 1all texts inside boxes should be in vertical direction and relatedness between the last three boxes (yellow) should be clarified by arrows direction like others.
Response: Corrected in revised manuscript.
- Table 2. title need to be moved to the next page
Response: Changes have been made in revised manuscript according to reviewer suggestion.
- Table 7, line 305 to be moved to the next page
Response: Corrected in revised version.
Good luck
Round 2
Reviewer 1 Report
There are still serious flaws in the study design. The parameters generated by bioinformatic software are not convincing without proper positive and negative controls. One example would be: the authors compared the antigenicity score before and after cystine modifications, however, neither of the constructs are validated in vivo, how would I know their antigenicity scores (0.69 or 0.67) indicates good antigenicity? At least the antigenicity score of a known immunogen that has been proven to work in vivo should be included as a comparison. And such controls should be presented in every prediction/stimulation.
Author Response
Response to Reviewer comments
We thank the Referee for spending time and interest in our work and for helpful comments that will greatly improve the manuscript. We have checked all the general and specific comments provided by the Referee and have made all the necessary changes according to his indications. Please refer to green highlighted sections in the revised manuscript.
Reviewer # 1
Comments and Suggestions for Authors
There are still serious flaws in the study design. The parameters generated by bioinformatic software are not convincing without proper positive and negative controls. One example would be: the authors compared the antigenicity score before and after cystine modifications, however, neither of the constructs are validated in vivo, how would I know their antigenicity scores (0.69 or 0.67) indicates good antigenicity? At least the antigenicity score of a known immunogen that has been proven to work in vivo should be included as a comparison. And such controls should be presented in every prediction/stimulation.
Response: Thank you for the valuable comment. The authors agree with the reviewer on using controls. In the revised manuscript, the authors have added OmpA protein from E. coli as positive control and know protective antigen against bacterial pathogens (doi.org/10.3389/fimmu.2020.01362). The comparative analysis is added at appropriate places and hopefully the reviewer will be convinced this time. Thank you.
Reviewer 2 Report
The revised format improved the quality of the manuscript. However little attention was given about the addition of the cholera toxin as adjuvant. Some comments regarding the importance of this molecule in the discussion should be raised.
Author Response
Response to Reviewer comments
We thank the Referee for spending time and interest in our work and for helpful comments that will greatly improve the manuscript. We have checked all the general and specific comments provided by the Referee and have made all the necessary changes according to his indications. Please refer to green highlighted sections in the revised manuscript.
Reviewer # 2
Comments and Suggestions for Authors
The revised format improved the quality of the manuscript. However little attention was given about the addition of the cholera toxin as an adjuvant. Some comments regarding the importance of this molecule in the discussion should be raised.
Response: Thank you for the valuable comment. The importance of cholera toxin as adjuvant molecule is added to revised manuscript discussion section.
Round 3
Author Response
Response to Reviewer comments
We thank the Referee for spending time and interest in our work and for helpful comments that will greatly improve the manuscript. We have checked all the general and specific comments provided by the Referee and have made all the necessary changes according to his indications.
Reviewer # 1
Response: Thank you for the valuable comments. The authors thoroughly revised each section of the manuscript and improved the text as per reviewer indications. The authors believe these revisions will be enough to make the paper acceptable for publication.